# Polysialylation in a DISC1 Mutant Mouse

**DOI:** 10.3390/ijms23095207

**Published:** 2022-05-06

**Authors:** Yuka Takahashi, Chikara Abe, Masaya Hane, Di Wu, Ken Kitajima, Chihiro Sato

**Affiliations:** 1Bioscience and Biotechnology Center, Nagoya University, Chikusa, Nagoya 464-8601, Japan; takahashi.yuka@i.mbox.nagoya-u.ac.jp (Y.T.); abe.c.nucoop@gmail.com (C.A.); mhane@agr.nagoya-u.ac.jp (M.H.); diwu@agr.nagoya-u.ac.jp (D.W.); kitajima@agr.nagoya-u.ac.jp (K.K.); 2Graduate School of Bioagricultural Sciences, Nagoya University, Chikusa, Nagoya 464-8601, Japan; 3Integrated Glyco-Biomedical Research Center (iGMED), Institute for Glyco-core Research (iGCORE), Nagoya University, Chikusa, Nagoya 464-8601, Japan

**Keywords:** schizophrenia, polysialic acid, genetic factor, environmental factor, DISC1, NCAM, polysialyltransferase, tail suspension test, acute stress, mental disorder

## Abstract

Schizophrenia is a serious psychiatric disorder that affects the social life of patients. Psychiatric disorders are caused by a complex combination of genetic (G) and environmental (E) factors. Polysialylation represents a unique posttranslational modification of a protein, and such changes in neural cell adhesion molecules (NCAMs) have been reported in postmortem brains from patients with psychiatric disorders. To understand the G × E effect on polysialylated NCAM expression, in this study, we performed precise measurements of polySia and NCAM using a disrupted-in-schizophrenia 1 (DISC1)-mutant mouse (G), a mouse model of schizophrenia, under acute stress conditions (E). This is the first study to reveal a lower number and smaller length of polySia in the suprachiasmatic nucleus of DISC1 mutants relative to those in wild-type (WT) mice. In addition, an analysis of polySia and NCAM responses to acute stress in five brain regions (olfactory bulb, prefrontal cortex, suprachiasmatic nucleus, amygdala, and hippocampus) revealed that the pattern of changes in these responses in WT mice and DISC1 mutants differed by region. These differences could indicate the vulnerability of DISC1 mutants to stress.

## 1. Introduction

Schizophrenia is a serious mental disorder with an incidence of approximately 1% of the global population. Patients experience various mental disorders (hyperactivity, emotional dullness, decreased motivation, impaired social behavior, and cognitive dysfunction), including realistic hallucinations and paranoia, which impact their social life [1]. However, the molecular mechanism of schizophrenia pathogenesis is not well understood, despite the availability of several symptomatic treatments for schizophrenia [2]. The largest genome-wide association study (GWAS) identified 108 genetic loci that were associated with schizophrenia [3]. Further, its onset is assumed to be the result of an interaction between multiple gene mutations. To date, only a few of these factors have been analyzed in detail and many factors remain unidentified. Moreover, understanding the complete mechanism of its pathogenesis is difficult, and increasing research on the role of each factor is important. Although results from twin, family, and adoption studies have strongly suggested the involvement of genetic factors (G) in the development of schizophrenia, environmental factors (E), such as time of birth, upbringing, infection, and drug use, have also been reported to be risk factors for psychiatric disorders including schizophrenia.

Polysialic acid (polySia/PSA) is the glycan associated with mental disorders [4]. It is a linear polymer of Sias with a degree of polymerization (DP) of 8–400 [5]. It is primarily present on the neural cell adhesion molecule (NCAM) in embryonic brains, and although it is found in the adult brain, its distribution is restricted [6,7,8]. The regions where polySia expression continues until adulthood include: the hippocampus (HIP) and olfactory bulb (OB), where adult neurogenesis remains ongoing; the suprachiasmatic nucleus (SCN), where the circadian clock is regulated; the amygdala (AMG), where emotion and fear are regulated; and the prefrontal cortex (PFC), where higher brain function is controlled. PolySia has an anti-adhesive effect on NCAM–NCAM or NCAM–other CAM interactions, leading to regulation of the intracellular space. PolySia also functions as a reservoir and it stores BDNF, FGF2, and dopamine to regulate its concentrations outside the cells [5,8].

To date, polySia has been shown to be associated with mental disorders, such as schizophrenia [4,9,10]. For example, from an anatomical point of view, in the postmortem brains of schizophrenia patients, the number of polySia-bearing neurons was found to decrease in the HIP [11]. In addition, a decrease in polySia levels has been reported in the PFC of patients [12]. The glycosyltransferase gene *ST8SIA2*, which is one of the genes encoding polysialyltransferases (polySTs) [13], has been reported to be related to mental disorders. Previously, a mutation at an open reading frame (ORF) of *ST8SIA2* in a schizophrenia patient [14] was analyzed biochemically [15]. This mutation was found to cause impairments in enzymatic activity, changes in the quantity and quality of reaction products (polySia), and altered molecule-binding properties and anti-adhesive functions of polySia [15,16,17,18]. GWASs revealed that several intronic SNPs (iSNPs) in *ST8SIA2*, such as iSNPs found in bipolar disorder patients, lead to increased amounts of polySia expression [19]; these biochemical results were consistent with those of another study revealing increased expression of polySia in the AMG of patients with bipolar disorder [20]. Interestingly, St8sia2-KO mice are used as a model to evaluate schizophrenia phenotypes, such as prepulse inhibition (PPI) [10], which were found to be the same as those found in schizophrenia patients. From an environmental perspective, polySia expression has been shown to be regulated by several environmental factors that are related to mental disorders. One of these factors is the anti-schizophrenia drug chlorpromazine. After mice were administered chlorpromazine, polySia expression was found to increase in the PFC of mice alone [21]. PFC was reported to be the location of this decrease in patients [12]. Further, when mice were exposed to acute stress, polySia expression was also found to decrease, especially in the PFC and OB, but this was increased in the SCN [22]. Interestingly, the effect of acute stress on the change in polySia was recovered within 3 h to 1 d, indicating that polySia expression is dynamically and highly regulated [22]. Accordingly, polySia expression is also highly regulated genetically and environmentally; however, such regulation has only been demonstrated independently. Thus, the expression of polySia under G × E conditions has not been analyzed. In this study, we opted to focus on disrupted-in-schizophrenia 1 (DISC1) [23], a well-known genetic factor not only in humans but also in mice [24], and evaluated polySia expression in five brain regions. The effects of stress on the environmental factors that affect polySia expression were also analyzed to understand the expression of polySia under G × E conditions.

## 2. Results

### 2.1. Design of Experiment

Several genetic and environmental factors are related to schizophrenia. For the genetic factor, DISC1 mutant mice with a point mutation (L100P) in the ORF of the gene [24] were used. These mice were used as a model for schizophrenia based on the evaluation of PPI and latent inhibition. DISC1 mutant mice have been used to analyze the mechanism of schizophrenia [25]. We selected tail suspension as an acute stress representing an environmental factor, which was established as a screening method for antidepressant drugs [26] (Figure 1A). Mice were divided into four groups for the experiment as follows: group 1 comprised wild-type (WT) mice that were not subjected to stress (WT, −); group 2 consisted of DISC1 mutant mice that were not subjected to stress (DISC1 mutant, −); group 3 comprised WT mice subjected to stress (WT, +); and group 4 consisted of DISC1 mutant mice subjected to stress (DISC1 mutant, +) (Figure 1B). Each group contained five 8-week-old male mice. The mice were housed in a room for at least 2 weeks for pre-habituation, and an experimental room for 1 h for habituation. After the experiments, the brain and blood of the mice were collected under anesthesia (Figure 1C). The five brain regions related to the expression of polySia/PSA (Figure 1D) were dissected, as described previously, and analyzed via further experiments.

### 2.2. Comparison between WT and DISC1 Mutant Mice: Evaluation of the Immobility Rate and Concentration of Corticosterone in Serum

The immobility rate of the mice was measured and it was increased among DISC1 mutant mice (Figure 2A). The concentrations of corticosterone in the serum derived from the four groups of mice (Figure 1B) were also determined using an enzyme-linked immunosorbent assay (ELISA). As shown in Figure 2B, the concentrations of stress markers were significantly increased in the serum of WT and DISC1 mutant mice (tail suspension [TS]-vs. TS+) after acute stress exposure, indicating that tail suspension markedly enhanced stress. There were no differences between WT and DISC1 mutant mice.

### 2.3. Evaluation of polySia (Glycan) Expression Based on 12E3, 735, and NCAM (Protein) Expression in the Brain

To evaluate polySia expression precisely, ELISA was performed using two different antibodies, 12E3 (Figure 3) and 735 (Figure 4). The 12E3 recognizes oligo/polySia with a nonreducing terminal end [5,27,28], whereas 735 recognizes polySia in its internal structure [5,28,29]. Both antibodies were used after endo-N-acylneuraminidase (endo-N) treatment. NCAM (Figure 5) protein levels were measured after endo-N treatment.

#### 2.3.1. OB

When polySia expression between group 1 (WT, −) and 2 (DISC1 mutant, −) was compared based on the 12E3 antibody, it was found to be lower in group 2 (Figure 3, OB). In contrast, upon comparison with 735, the expression pattern did not change (Figure 4, OB). When groups 1 (WT, −) and 3 (WT, +) were compared, the expression of polySia tended to decrease (Figure 4, OB). This result is consistent with that of a previous study [22]. The same effect was observed when groups 2 (DISC1 mutant, −) and 4 (DISC1 mutant, +) were compared (Figure 4, OB). NCAM expression tended to increase when groups 1 (WT, −) and 2 (DISC1 mutant, −) were compared (Figure 5, OB). However, NCAM did not increase after TS.

#### 2.3.2. PFC

A significant decrease in polySia expression in the PFC was only observed when evaluated with the 735 antibody (Figure 4, PFC). Notably, the amounts of polySia and NCAM in groups 1 (WT, −) and 2 (DISC1 mutant, −) did not change. However, after TS, polySia decreased as compared to that found in groups 1 (WT, −) and 3 (WT, +). This result is consistent with that of a previous study [22]. When DISC1 mutant mice were evaluated, a decrease in polySia was observed, and the decreasing ratio was almost the same (Figure 4. PFC).

#### 2.3.3. SCN

When polySia expression was compared between groups 1 (WT, −) and 2 (DISC1 mutant, −), the expression in DISC1 mutant mice was found to significantly decrease using 12E3 and 735 antibodies (Figure 3 and Figure 4, SCN). At the same time, NCAM expression tended to decrease in DISC1 mutant mice (Figure 5, SCN). These findings indicate that DISC1 mutant mice have a low level of polySia in their SCNs. PolySia expression increased in groups 1 (WT, −) and 3 (WT, +) after TS treatment (Figure 3 and Figure 4, SCN). This was also observed when DISC1 mice from groups 2 (DISC1 mutant, −) and 4 (DISC1 mutant, +) were compared. Notably, this tendency was not observed in WT mice, which might be due to the exclusion points; however, an increase in NCAMs between groups 2 (DISC1 mutant, −) and 4 (DISC1 mutant, +) was observed.

#### 2.3.4. AMG

Based on the comparison of groups 1 (WT, −) and 2 (DISC1 mutant, −) using 12E3 and 735 antibodies, the expression of polySia and NCAM did not change (Figure 3, Figure 4 and Figure 5, AMG). However, when polySia expression between groups 1 (WT, −) and 3 (WT, +) was compared, it was found to be increased (Figure 3 and Figure 4, AMG). These increases were not observed when groups 2 (DISC1 mutant, -) and 4 (DISC1 mutant, +) were compared. Overall, the AMG was found to display different phenotypes in terms of the different genetic factors under environmental conditions. There was no change in NCAM expression (Figure 5, AMG).

#### 2.3.5. HIP

When polySia expression was compared between groups 1 (WT, −) and 2 (DISC1 mutant, −) using 12E3, 735, and NCAM antibodies, no change was observed (Figure 3, Figure 4 and Figure 5, HIP). However, using 12E3, a decrease in polySia expression was observed between groups 1 (WT, −) and 3 (WT, +). A similar decrease was observed in DISC1 mutant mice between groups 2 (DISC1 mutant, −) and 4 (DISC1 mutant, +), and the decreasing ratio was the same as that found in WT mice (Figure 4. PFC). No significant changes were observed using the 735 and NCAM antibodies (Figure 4 and Figure 5, HIP).

## 3. Discussion

### 3.1. Immobility Ratio and Concentration of Corticosterone

Corticosterone, also known as a corticosteroid, is an adrenal corticosteroid involved in metabolism and immune responses in rodents. The cortisol levels in humans are already known. In fact, their secretion and serum levels have been found to increase when animals are subjected to stressful stimuli. Therefore, corticosterone in rodents and cortisol in humans are used as biomarkers of stress [30]. As the concentrations of corticosterone in the serum of WT and DISC1 mutant mice increased two-fold after acute stress exposure (Figure 2B), TS was determined to be useful for both the genetically different mice. DISC1 is known to be involved in stress responses and has been reported to regulate hormone secretion upstream of the hypothalamic-pituitary-adrenal (HPA) axis, which responds to stress by regulating neuroendocrine and autonomic responses [31]. Stress exposure induces the secretion of corticotropin-releasing hormone (CRH) from the hypothalamus, which stimulates the secretion of adrenocorticotropic hormone (ACTH) from the anterior pituitary gland. ACTH stimulates the adrenal cortex and promotes the secretion of stress hormones [32]. DISC1 is involved in CRH release from the hypothalamus upstream of the HPA axis, and studies on zebrafish have revealed that DISC1 mutations cause CRH dysregulation, resulting in the failure to appropriately increase cortisol levels under stress and causing abnormal social behavior. A significant difference was not found between WT and DISC1 mutant mice in this study, under normal and stress conditions.

### 3.2. Analysis of polySia and NCAM Expression in Five Different Brain Regions

#### 3.2.1. OB

The OB plays a role in processing and transmitting information on odor molecules detected in the nasal cavity. Rats in which the OB is removed show changes similar to those found in depressed patients, and these animals have been used in various studies as a model organism for depression [33]. The relationship between polySia and mental disorders has not been reported; however, the decreased volume of the OB in NCAM-KO mice [34] is similar to that observed in schizophrenia patients [35,36].

A previous study, which analyzed the change in polySia expression in the brain by exposing mice to acute stress [22], revealed a significant decrease in polySia after TS using the 735 antibody, similar to the observations of the present study (Figure 6, 735, OB). There was no significant difference in the expression of polySia and NCAM, as stress response markers, between the WT and DISC1 mutant animals. In the present study, polySia in the OB was affected by acute stress (E); however, the stress response was not found to be altered by genetic factors (G).

#### 3.2.2. PFC

The PFC is responsible for processing information from various stimuli and is involved in cognitive function and decision-making. Further, it is critical for motivation and attention. Damage to it, or its removal, results in increased impulsivity and difficulties in making arrangements [37]. The expression of polySia has been reported to decrease in the PFC of schizophrenic brains [12]. In this experiment, we analyzed the relationship between G and E with respect to polySia expression. TS did not result in a significant difference in polySia expression in WT animals using the 12E3 antibody; however, a significant decrease was found using 735 (Figure 7, 735, PFC), aligning with the results of previous studies [22]. There was no significant difference in the ratio of the polySia decrease as a stress response between WT and DISC1 mutant mice (Figure 4, PFC). Thus, in the present system, polySia in the PFC was affected by acute stress (E), whereas the stress response was not altered by genetic factors (G). In contrast, neither WT nor DISC1 mutant mice showed stress-induced alterations in NCAMs, suggesting that the reduction in polySia expression due to acute stress is not regulated by the amount of NCAM present but rather by a single regulatory mechanism for polySia degradation, as revealed in a previous study [22].

#### 3.2.3. SCN

The SCN, a part of the hypothalamus, is known as the core of homeostasis, especially with regard to the biological clock [38]. One of the symptoms of patients with schizophrenia and other psychiatric disorders is sleep disturbances [39,40]. Further, DISC1 is known to regulate sleep/wake states via D2 dopamine receptors [41]. Experiments using a transgenic mouse model expressing full-length human DISC1 revealed longer wakefulness in transgenic mice than in WT controls [42]. The relationship between polySia and SCN has been well-characterized [43]. Further, polySia has been revealed to be important for regulating the circadian rhythm in experiments involving its removal from the SCN.

The number of polySia molecules determined using 12E3 and the length of polySia examined using 735 were 67% and 59%, respectively, in DISC1 mutant mice under normal conditions, compared to 100% observed in WT animals (Figure 3 and Figure 4, SCN; Figure 6). In a mouse model of circadian clock impairment (NCAM-KO mice and DISC1-KO mice), few polySia molecules were found in the SCN, which controls the circadian rhythm, suggesting that its presence in the SCN is associated with the sleep disturbances associated with mental disorders, such as schizophrenia.

As mentioned, DISC1 has been shown to be involved in the stress response. Further, individuals with mutations in DISC1 have been found to have altered pituitary stimulating hormone (CRH) levels and display abnormal social behavior as their corticosterone levels do not increase under stress [32]. Although corticosterone levels did not differ between WT and DISC1 mutant animals in the 7 min TS system in the present study, the low amount of polySia in the hypothalamic region, which is the origin of the HPA axis, of DISC1 mutant mice may indicate some dysfunction with respect to the stress response. Furthermore, a comparison of the increasing/decreasing patterns of polySia and NCAM in the four groups suggests that they are regulated in a generally synchronous manner. Notably, polySia-NCAM, which might be a soluble type of polySia-NCAM, is exocytosed from the cell and regulates transcription factors. Thus, polySia-NCAM cleaved by metalloproteases might be involved in the circadian system.

#### 3.2.4. AMG

The AMG is one of the regions responsible for emotion and is involved in fear conditioning. Studies with rhesus monkeys have revealed that damage to the temporal lobe, including the AMG, causes profound social and emotional deficits. A relationship between polySia and the AMG was found in patients with BD. In these patients, the amount of polySia in the AMG, in the lateral nucleus, was also found to be increased. In this study, acute stress (TS) significantly increased polySia in the AMG of WT mice; however, such an increase was not observed in the DISC1 mutant mice (Figure 3 and Figure 4, AMG). To the best of our knowledge, this is the first study to reveal the role of polySia under G × E conditions, suggesting its involvement in the resulting pathogenic mechanism of mental disorders (Figure 6, AMG, 12E3/735, WT/DISC1). However, as neither WT nor DISC1 mutant mice showed stress-induced changes in NCAMs (Figure 6, AMG, NCAM), the increase in polySia in response to acute stress was not regulated by the amount of NCAM present, but rather by the regulatory mechanism of polySia alone (Figure 5, AMG).

#### 3.2.5. HIP

The HIP is considered the region responsible for memory. In fact, removal of the human HIP was found to induce severe progressive memory impairment [44]. Prolonged corticosterone exposure owing to chronic stress was also reported to damage hippocampal cells [45], and chronic stress increases polySia in the HIP [46,47]. Chronic stress is defined as the accumulation of stress and enhances the concentration of corticosterone. In turn corticosterone could destroy the tissues. In this study, we focused on acute stress alone because chronic stress represents a prolonged and complex stimulus.

A decrease in polySia in the HIP was reported in schizophrenia patients. Notably, the effect of antidepressant drugs on diseases is related to the amount of polySia. Interestingly, working memory and long term potentiation (LTP) or long term depression (LTD) were found to be impaired in polyST-KO mice. In this study, we analyzed the change in polySia in the HIP under G × E conditions. Acute stress reduced the number of polySia molecules in both WT and DISC1 mutant mice based on the 12E3 antibody (Figure 6, 12E3, WT and DISC1); however, no significant difference was found with 735. These results suggest that, in the HIP, the effect of acute stress is reflected by the number of polySia molecules rather than in its length (Figure 3 and Figure 4, HIP). Interestingly, in this study, acute stress, unlike chronic stress, was found to decrease polySia. As an increase in the pulse of corticosterone was observed; however, we already had confirmed that the corticosterone pulse increase did not affect polySia expression in the five brain regions. Neither WT nor DISC1 mutant NCAM was altered by stress, suggesting that the decrease in polySia due to acute stress is not regulated by the amount of NCAM present. Therefore, a polySia metabolism-related mechanism such as sialidase Neu1, as demonstrated in a previous study, might be involved.

### 3.3. PolySia Expression and DISC1

The *DISC1* gene was found in a Scottish family with a translocation between chromosomes 1 and 11 and multiple psychiatric disorders [48]. This translocation is assumed to disrupt two genes on chromosome 1, one of which is *DISC1*. Several molecules, such as GSK3β, NDEL1, girdin, and PDE4, have been reported in such interactomes, including microtubule-binding molecules and synaptic signaling molecules [49]. Typical functions of these molecules are assumed to include the neurodevelopment of the neocortex and HIP, and synaptic regulation. As DISC1 has many binding proteins and regulates these molecules, it is considered to function in various nervous system activities, such as neurodevelopment and synaptic regulation [50]. Intracellularly, DISC1 localizes to centrosomes, microtubules, and growth cones [51], suggesting that it functions during neurogenesis, such as in neuronal proliferation and migration, dendrite formation, and axon elongation. Indeed, the inhibition of DISC1 function during development results in reduced neuronal proliferation and differentiation into premature neurons, delayed neuronal migration in the neocortex and HIP, impaired dendrite formation, and impaired axon elongation [50]. However, the inhibition of DISC1 function in the adult hippocampal dentate gyrus has been reported to result in excessive neuronal migration [52]. Some of the dysfunctions were found to overlap with those of polySia [53]. In this study, we found different expression levels of polySia in DISC1 mutant mice, especially in the SCN, indicating an interaction between polySia and DISC1 molecules.

In summary, we analyzed the effect of G × E on polySia and NCAM expression and found that the change in polySia expression was region-specific. In particular, polySia in the SCN was found to be sensitive to genetic factors, such as DISC1. Moreover, polySia in the PFC and SCN, evaluated using the 735 antibody, and polySia in the HIP, evaluated using 12E3, were sensitive to environmental factors. The AMG was the only region affected by G × E conditions.

## 4. Materials and Methods

### 4.1. Materials

The corticosterone ELISA kit was purchased from Cayman (Ann Arbor, MI, USA). Bovine serum albumin (BSA) was purchased from Sigma-Aldrich (St. Louis, MO, USA). The anti-polySia antibody, 12E3, which recognizes the oligo/polyNeu5Ac structure (DP ≥ 5), was generously provided by Dr. Tatsunori Seki (Tokyo Medical University, Japan). The 735 antibody, which recognizes the polyNeu5Ac structure (DP ≥ 11), and endo-N, which cleaves the oligo/polySia structure (DP ≥ 5), were purified, as described previously [28]. The anti-NCAM antibody was purchased from R&D Systems (Minneapolis, MN, USA). POD-labeled anti-mouse IgG + IgM and anti-rabbit IgG were purchased from American Qualex (San Clemente, CA, USA). Phenylmethylsulfonyl fluoride was purchased from Wako (Osaka, Japan). Polyvinylidene difluoride membranes were procured from Millipore (Billerica, MA, USA). Nunc™ Multisorp surface immunoplates were purchased from Thermo Fisher Scientific (Waltham, MA, USA).

### 4.2. Animals and Ethics Statement

Mice (B6(D2)-Disc1 < Rgsc1390 >/Rod) were obtained from RIKEN (Saitama, Japan), and DISC1 mutant mice (8 W, male) were maintained in our laboratory. The C57/BL6J (male, 8 weeks of age) were obtained from Chubu Kagaku Shizai (Nagoya, Japan) and maintained in a controlled environment (23 ± 2 °C and 50 ± 10% humidity, 12:12 light/dark cycle) with food and water available ad libitum. At least 2 weeks prior to the experiments, the mice were allowed to habituate in our facility. All procedures were approved by the Animal Care and Use Committee of Nagoya University (permit number 2016031249). Every effort was made to minimize the number of animals used and their suffering.

### 4.3. TS Test

The mice were maintained for at least 2 weeks. In addition, mice were maintained in the experimental room for 1 h for habituation to the environment. Following this, they were suspended 30 cm above the floor by their tails for 7 min. The immobility time of mice for 6 min, excluding the first 1 min, was measured [22].

### 4.4. Sample Preparation

Mice were euthanized via CO_2_ administration immediately after TST for the TS+ (Stress+) group or after habituation in the experimental room for the TS−(Stress−) group. The cerebrum was surgically removed, and blood samples were retrieved from the mice. The OB was also dissected and collected. The remaining cerebrum was soaked in 1% agarose in PBS. After hardening, the cerebrum was clonally sliced into 500 μm sections using the Super MICROSLICER (DOSAKA EM, Kyoto, Japan) (Figure 1C). The PFC, SCN, AMG, and HIP were extracted from the sections as described previously [22]. The five regions of the brain were separated and homogenized in a lysis buffer (1% triton-X100, 1 mM phenylmethanesulfonyl fluoride, protease inhibitors: 1 μg/mL aprotinin, 1 μg/mL leupeptin, 1 μg/mL pepstatin, 2 μg/mL antipain, 10 μg/mL benzamidine, 1 mM EDTA, 50 mM NaF, 10 mM β-glycerophosphate, 10 mM sodium pyrophosphate, and 1 mM sodium o-vanadate in PBS). Homogenates were incubated on ice for 1 h and centrifuged at 10,000 rpm for 15 min at 4 °C. The supernatant was then collected, and protein concentrations were determined using the BCA assay.

### 4.5. Corticosterone Quantification

To measure the degree of stress, blood was collected after surgery, incubated for 1 h at 25 °C, and stored overnight at 4 °C. Following this, it was centrifuged at 3000 rpm for 10 min, and serum was collected. Serum was diluted 500-fold using the ELISA buffer. The diluted serum (5 μL) was dispensed in a well containing 50 μL of anti-corticosterone antibody and 50 μL of corticosterone-acetylcholinesterase. The plates were incubated overnight at 4 °C. The liquid was subsequently removed, and the well was rinsed five times with wash buffer. Reagents containing acetylcholine and 5,5′-dithio-bis-(2-nitrobenzoic acid) were added (50 μL per well) and incubated for 90 min at 25 °C. The absorbance was measured at 412 nm.

### 4.6. Quantification of polySia and NCAM Using ELISA

Samples were adjusted to 5 μg/mL (as BSA) with PBS (the concentration of triton X-100 was less than 0.03%). Thereafter, 50 μL of the solution was absorbed onto a 96-well immunoplate at 37 °C for 2 h. After washing with PBS, 100 μl of 2% BSA/PBS blocking solution was added and incubated at 37 °C for 1 h. After another wash with PBS, the wells were incubated with 50 μL of endo-N solution (35 μU/mL) to specifically cleave polySia or with 50 μL of PBS at 37 °C or 2 h. Following another PBS wash, 50 μL of primary antibody (12E3 (2 μg/mL), 735 (1/20 of culture medium), or anti-NCAM (2 μg/mL)) was added and incubated at 4 °C for 16 h. After washing with PBST, 50 μL of POD-labeled secondary antibody (anti-IgG + M (1 μg/mL) or anti-rat IgG (2 μg/mL)) was added and incubated at 37 °C for 1 h. Following washing with PBST, 100 μL of the substrate solution (0.1 M tris-HCl (pH 6.8), containing 0.05% *o*-phenylenediamine, 3% H_2_O_2_ for 12E3 or TMB solution for 735 and anti-NCAM antibodies) was added. The reaction was stopped by adding 100 μL of 2 N H_2_SO_4_. The absorbance was then measured at A490 for *o*-phenylenediamine or A450 for the TMB solution. All data were analyzed in triplicate for each sample.

### 4.7. Data Analysis

All values are expressed as the mean ± SEM (n is indicated), and *p*-values were evaluated using the Student’s *t*-test.

## Figures and Tables

**Figure 1 ijms-23-05207-f001:**
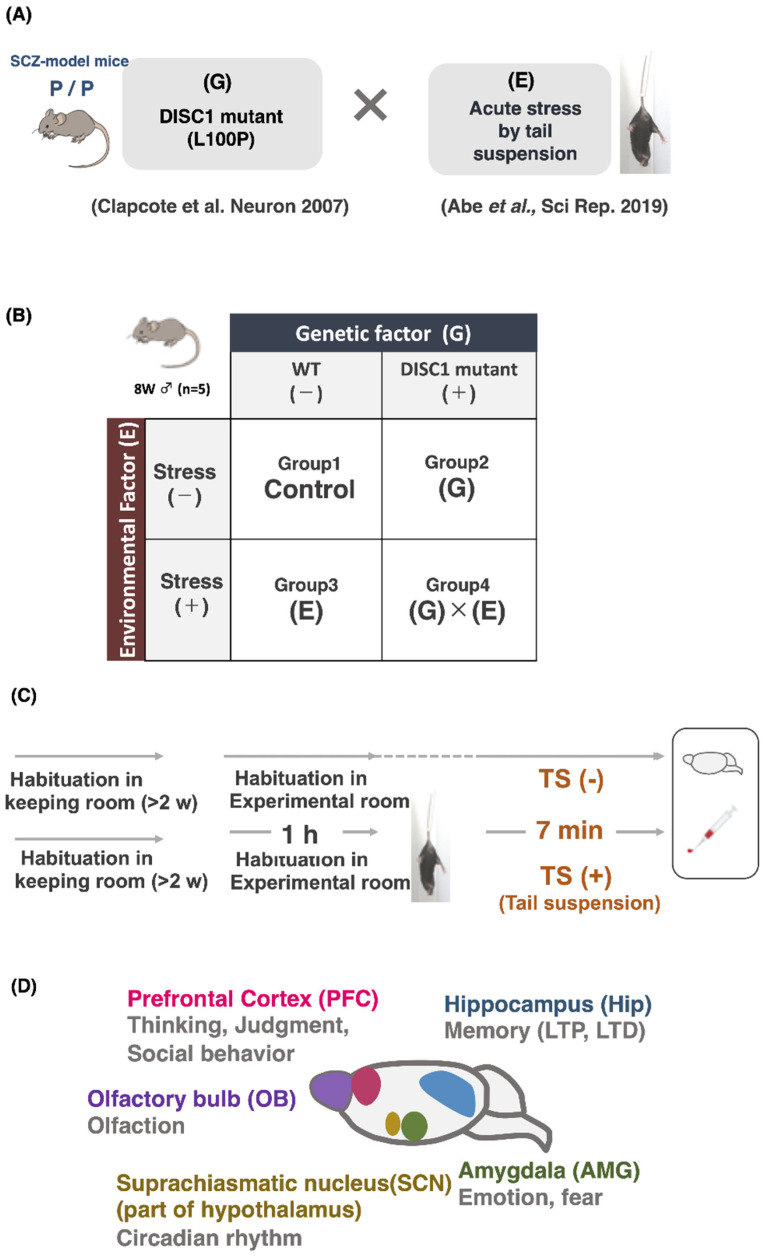
Experimental design of this study. (**A**) Interaction between genetic factor (G) and environmental factor (E), G × E. In this study, a schizophrenia mouse model (DISC1 mutant mice, L100P) was used as a contributing genetic factor. The environmental factor was established by exposing the mice to an acute stress condition, tail suspension. (**B**) Experimental design. Mice were divided into the control, G, E, and G × E groups, with five male mice per group. (**C**) Tail suspension test. Before this, pre-habituated mice (at least 2 weeks before experiment) were habituated for 1 h in the experimental room. Immediately after the end of the experiment, serum collection and brain extraction were performed. (**D**) Brain regions used in this study. The extraction was performed as described by Abe et al.

**Figure 2 ijms-23-05207-f002:**
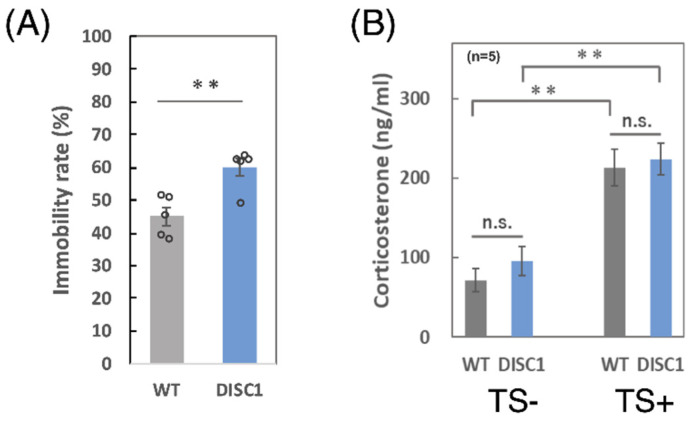
Comparison between WT and DISC1 mutant mice. (**A**) Immobility ratio. The immobility ratio of the mice (*n* = 5, 8 weeks, male) was measured during a tail suspension test. Error bars indicate the SEM (**: *p* < 0.01). (**B**) Concentration of corticosterone. The concentration of corticosterone, which is a marker of stress exposure, in serum was analyzed in the four groups of mice. DISC1 indicates DISC1 mutant mice and TS indicates the tail suspension. The concentration of corticosterone was determined using ELISA. Error bars indicate the SEM (**: *p* < 0.01).

**Figure 3 ijms-23-05207-f003:**
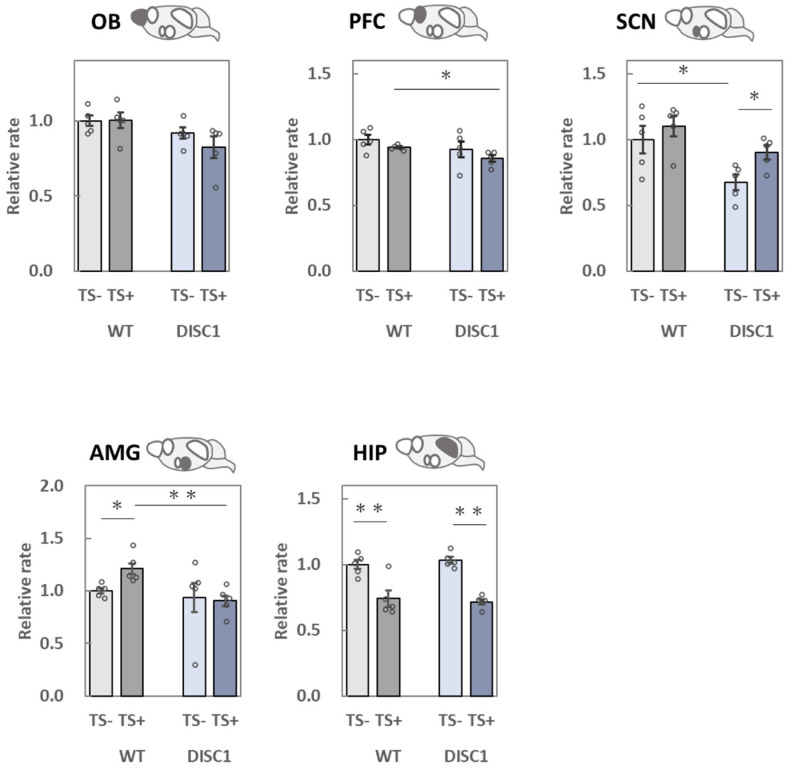
Measurement of the amounts of polySia by ELISA (12E3 antibody). PolySia expression levels in the five brain regions collected from the four groups of mice were analyzed. The samples were added to the wells of a 96-well plate (250 ng as protein), and the amounts of polySia were determined using the 12E3 anti-polySia antibody. The same sample was treated with endo-N, and subtraction was performed for specific polySia measurements. All data points represent averages of the three assays. The average in group 1 [WT, TS-] was set to 1. Error bars indicate the SEM (*n* = 5). OB: olfactory bulb, PFC: prefrontal cortex, SCN: suprachiasmatic nucleus, AMG: amygdala, HIP: hippocampus. *: *p* < 0.05, **: *p* < 0.01.

**Figure 4 ijms-23-05207-f004:**
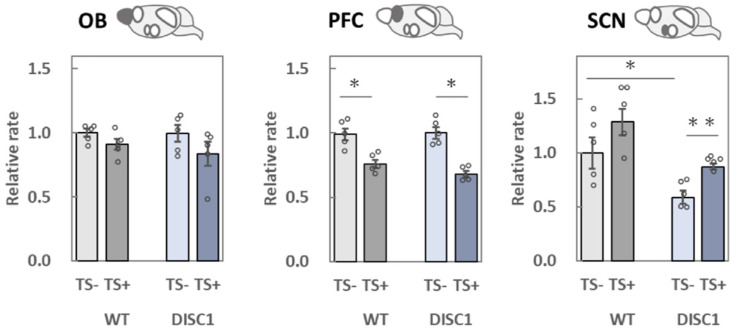
Measurement of polySia levels by ELISA (735 antibody). PolySia expression levels in the five brain regions collected from the four groups of mice were analyzed. The samples were added to the wells of a 96-well plate (250 ng as protein), and the amounts of polySia were determined using the 735 anti-polySia antibody. The same sample was treated with endo-N, and subtraction was performed for specific polySia measurements. All data points represent averages of the three assays. The average in group 1 [WT, TS-] was set to 1. Error bars indicate the SEM (*n* = 5). OB: olfactory bulb, PFC: prefrontal cortex, SCN: suprachiasmatic nucleus, AMG: amygdala, HIP: hippocampus. *: *p* < 0.05, **: *p* < 0.01.

**Figure 5 ijms-23-05207-f005:**
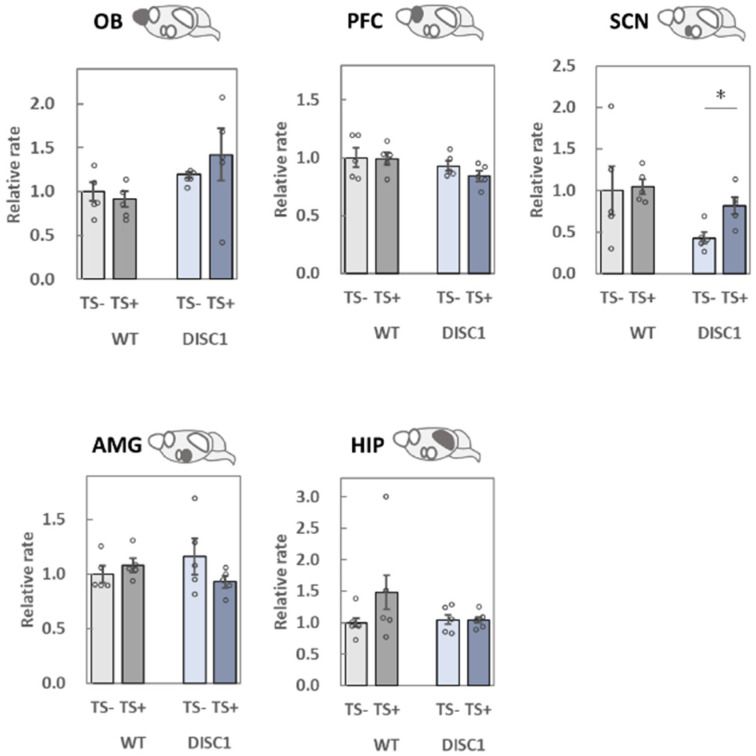
Measurement of NCAM levels using ELISA. NCAM expression levels in the five brain regions collected from the four groups of mice were analyzed. The samples were added to the wells of a 96-well plate (250 ng as protein) and treated with endo-N to determine the NCAM protein levels. NCAM was measured using the anti-CD56 antibody. All data points (each mouse) represent the average of the three assays. The average in group 1 [WT, TS-] was set to 1. Error bars indicate the SEM (*n* = 5). OB: olfactory bulb, PFC: prefrontal cortex, SCN: suprachiasmatic nucleus, AMG: amygdala, HIP: hippocampus. *: *p* < 0.05.

**Figure 6 ijms-23-05207-f006:**
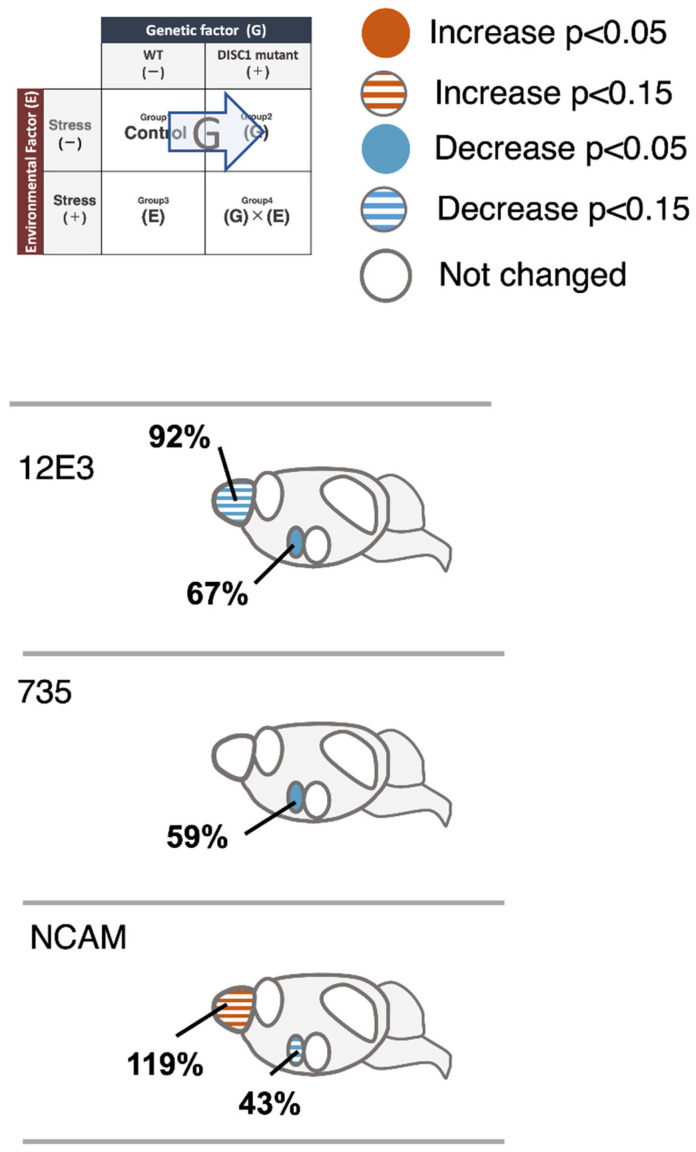
Effect of a genetic factor (DISC1 mutant) on the expression of polySia and NCAM. Heat map of changes observed using the 12E3, 735, and NCAM antibodies. [WT, TS−] (group 1) was set to 100%.

**Figure 7 ijms-23-05207-f007:**
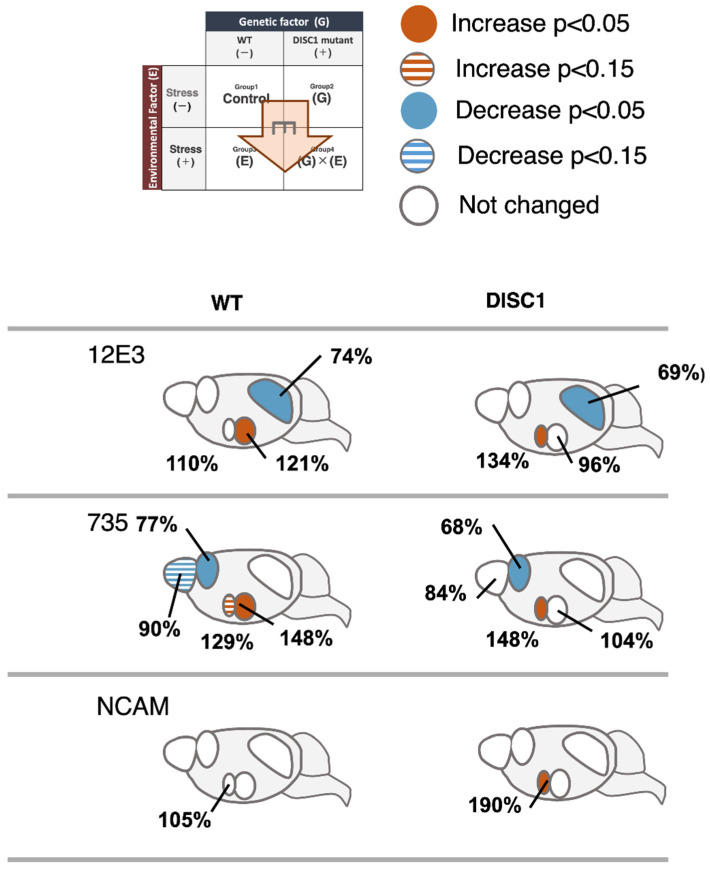
Effect of an environmental factor (acute stress) on the expression of polySia and NCAM. Heat map of changes observed using the 12E3, 735, and NCAM antibodies. [WT, TS−] (group 1) was set to 100%.

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
