# Peer review of "Polysialylation in a DISC1 Mutant Mouse"

_ijms, 2022, doi:10.3390/ijms23095207_

Round 1
Reviewer 1 Report
Schizophrenia is a neurodevelopmental disorder that affects people thinking and behaving. Polysialic acid (polySia) plays a significant role in regulating neuronal migration and differentiation during nervous system development. To understand how polySia contribute to schizophrenia, Takahashi and colleagues investigated polySia expression of critical brain regions and nuclei in a schizophrenia mouse model with DISC1 mutations. The author demonstrated that the number and length of polySia vary greatly from genetic to environmental factors in DISC1 mutant mouse model. The study can be improved with the following specific points being appropriately addressed.
- polySia and NCAM expression levels increased after tail suspension in SCN. It is interesting that DISC1 mice showed polySia and NCAM levels closed to WT. Does it mean polySia has been restored after stress stimuli? Do you have other evidence that SCN is affected by stress stimuli that associates with polySia? Can you discuss about the observation?
- The corticosterone measurement does not show significant difference between WT and DISC1 mice. It is hard to draw the conclusion that DISC1 mice are sensitive for stress according to corticosterone results. Please provide other results to support your claim.
- PFC and HIP are sensitive to different polySia antibodies with environmental stimuli. What is the functional difference between long and short polySia in cellular adhesion? It is critical for understanding distinct polySia levels in different brain regions. Do you measure the mRNA levels of ST8SiaII and ST8SiaIV?
- AMG shows deficits in stress response in DISC1 mice with both 12E3 and 735 antibodies. Whereas, PFC only shows significant difference with 12E3 antibody. Does it mean short polySia express higher in PFC? Does it play an important row in cortical plate development?
- There are a number of typo and grammar glitches, for example line 117, “ and the it”, line 65, needs definition of “iSNP”.
Reviewer 2 Report
First of all, the article requires major English editing where there are several wrong grammar and wrong word usage along with run-on sentences rendering the work nonunderstandable.
the article aims to assess the posttranslational modification of Polysialylation in DISC1-mutant mice. the authors do not describe what is Polysialylation structure and what is the value of studying and the aim of the work.
is it aimed to investigate a biomarker level or assess mechanistic changes; neither is described.
the work is descriptive with only ELISA and one behavioral testing of tail suspension with n=5.
this study is underpowered with a low number of mice, the changes of NCAM and its PTMshould be shown by Western blotting,. In addition, the authors sacrificed the mice immediately after experimentation which doesn't make sense as Post translational modifications of proteins would take time to change . !!!!!!
thus the work suffers lots of major weaknesses that cant be corrected by additional experimentations
Reviewer 3 Report
The authors show the level of polySia in WT and DISC1 mutant mice under normal and stressed condition in different regions of the brain. I have a couple of questions:
- In line 124, the authors claim that DISC1 mutant mice tend to have higher corticosterone under normal condition but the results show that the trend is not significant. Could the authors comment on that? Were other assays conducted to prove otherwise?
- In line 314, in AMG, the authors say that acute stress increases polySia in WT mice, but the results doesn't show any significant change in the WT under normal and stressed condition. However ,DISC1 mutant mice showed decrease under stress condition than normal conditions. There is also no significant difference when compared between WT and mutant mice under stress condition. Could the authors comment on their statement?
- All the results should have statistical significance, otherwise it is difficult to comment.
Round 2
Reviewer 1 Report
The manuscript need English language polishing to remove a number of grammar glitches.
The authors claimed that they tested mRNA levels of ST8SIA2 and ST8SIA4 without providing any evidence.
Please add discussion about functional differences between long and short polySias to the manuscript.
Author Response
We asked this manuscript to check the English grammar two times. Could you point out the places we should edit.
We added the data for real-time PCR.

Reviewer 2 Report
Thank you for the response,
Please add the details of the antibodies companies and concentrations as well as the ELISA companies.
Author Response
We had described the antibody origin, and the concentrations in detail. We established the ELISA methods by ourselves because the Endo-N based ELISA-method is not available from company products. The company name of the antibodies used in our study were all described in original and this manuscript.